# Characterization of neutralizing antibody with prophylactic and therapeutic efficacy against SARS-CoV-2 in rhesus monkeys

Shuang Wang[1,2,8], Yun Peng[3,8], Rongjuan Wang[1,2,8], Shasha Jiao[1,2,8], Min Wang[1,8], Weijin Huang [4,8], Chao Shan [5], Wen Jiang[1], Zepeng Li[1], Chunying Gu[1], Ben Chen[1], Xue Hu[5], Yanfeng Yao[3], Juan Min[6], Huajun Zhang[3], Ying Chen[3], Ge Gao[3], Peipei Tang[1], Gang Li[1], An Wang[1], Lan Wang[4], Jinchao Zhang [1], Shuo Chen [7✉], Xun Gui [1✉], Zhiming Yuan [3✉] & Datao Liu [1✉]

Efficacious interventions are urgently needed for the treatment of COVID-19. Here, we report a monoclonal antibody (mAb), MW05, with SARS-CoV-2 neutralizing activity by disrupting the interaction of receptor binding domain (RBD) with angiotensin-converting enzyme 2 (ACE2) receptor. Crosslinking of Fc with FcγRIIB mediates antibody-dependent enhancement (ADE) activity by MW05. This activity is eliminated by introducing the LALA mutation to the Fc region (MW05/LALA). Potent prophylactic and therapeutic effects against SARS-CoV-2 are observed in rhesus monkeys. A single dose of MW05/LALA blocks infection of SARS-CoV-2 in prophylactic treatment and clears SARS-CoV-2 in three days in a therapeutic treatment setting. These results pave the way for the development of MW05/LALA as an antiviral strategy for COVID-19.

[1] Mabwell (Shanghai) Bioscience Co., Ltd, 201210 Shanghai, China. [2] Beijing Kohnoor Science & Technology Co., Ltd, 102206 Beijing, China. [3] Center for Biosafety Mega-Science, Wuhan Institute of Virology, Chinese Academy of Sciences, 430071 Wuhan, Hubei, China. [4] Key Laboratory of the Ministry of Health for Research on Quality and Standardization of Biotech Products, National Institutes for Food and Drug Control, 100050 Beijing, China. [5] State Key Laboratory of Virology, Wuhan Institute of Virology, Chinese Academy of Sciences, 430071 Wuhan, Hubei, China. [6] Wuhan Institute of Virology, Chinese Academy of Sciences, 430071 Wuhan, Hubei, China. [7] Ludwig Cancer Research, Nuffield Department of Medicine, University of Oxford, Oxford OX3 7DQ, UK. [8] These authors contributed equally: Shuang Wang, Yun Peng, Rongjuan Wang, Shasha Jiao, Min Wang, Weijin Huang. ✉email: ericshuochen@gmail.com; xun.gui@mabwell.com; yzm@wh.iov.cn; datao.liu@mabwell.com

C OVID-19, caused by SARS-CoV-2, is currently spreading globally, threatening human health and economic development[1,2]. As of 27 July 2020, COVID-19 has resulted in more than 16 million infections and 647,784 deaths. Although multiple clinical trials are ongoing to evaluate repurposing anti-viral and anti-inflammatory agents, no specific treatment against SARS-CoV-2 has been approved since the worldwide outbreak began 6 months ago[3]. Treatments using plasma from convalescent COVID-19 patients have shown clear clinical improvement of both mild and severe cases of COVID-19, indicating that passive administration of neutralizing mAbs could have a major impact on controlling the SARS-CoV-2 pandemic by providing immediate protection[4,5]. During the SARS and Middle East respiratory syndrome coronavirus (MERS-CoV) outbreaks, a number of neutralizing mAbs were developed and proved their potential therapeutic uses for the treatment of coronavirus infections[6,7]. Neutralizing antibodies for Ebola virus, mAb114 and REGN-EB3, are other encouraging examples that using antibody-based therapy can be effective during an infectious disease outbreak[8–10].

The spike (S) protein on the surface of SARS-CoV-2 is the major molecular determinant for viral attachment, membrane fusion, and entry into host cells. Therefore, this protein is the main target for development of neutralizing antibodies and vaccines. Previous studies revealed that a large number of antibodies targeting the receptor-binding domain (RBD) of either SARS-CoV or MERS-CoV showed potent neutralizing activities by disrupting the interaction of spike protein with receptors on host cells[11–13]. Screening of RBD-targeting antibodies is the most straightforward way to generate SARS-CoV-2-neutralizing antibodies.

Here we show the identification and characterization of a SARS-CoV-2 RBD-targeting mAb, MW05, with high neutralization activity by disrupting the interaction of RBD with ACE2 receptor. FcγRIIB was confirmed involving the antibody-dependent enhancement (ADE) of SARS-CoV-2 infection mediated by MW05. Introducing of LALA mutation to the Fc region (MW05/LALA) completely abolished the ADE activity of MW05. Further, potent prophylactic and therapeutic effects against SARS-CoV-2 were observed in rhesus monkeys. These results support the development of MW05/LALA for combating COVID-19. MW05/LALA will enter phase 2 clinical trial soon.

## Results

**Identification and characterization of MW05 and MW07.** To obtain fully human SARS-CoV-2 neutralizing mAbs, we first generated SARS-CoV-2 RBD recombinant protein. We used this protein as bait to isolate specific memory B cells from peripheral blood mononuclear cells of a COVID-19 convalescent patient. We then used a single B cell cloning strategy to amplify the coding sequence of variable regions of IgG antibodies from individual B cells and insert them into human IgG1 vectors for recombinant antibody expression[14]. A large panel of SARS-CoV-2 RBD-binding mAbs were generated and characterized. Two mAbs, MW05 and MW07, showed high RBD-binding abilities and strong RBD/ACE2-disrupting activities in enzyme-linked immunosorbent assay (ELISA). IC$_{50}$ was determined to be 0.054 μg/ml for MW05 and 0.037 μg/ml for MW07 (Fig. 1a–c). MW05 and MW07 could also block the interaction of SARS-CoV-2 RBD with ACE2-overexpressing HEK293 cells (Supplementary Fig. 1). Fluorescence-activated cell sorting (FACS) analysis showed that both mAbs could specifically bind to SARS-CoV-2 S protein expressed on HEK293 cells (Fig. 1d). The dissociation constants ($K_d$) of MW05 and MW07 binding to SARS-CoV-2 S1 recombinant protein were measured by a surface plasmon resonance (SPR) assays. $K_d$ was 0.403 nM for MW05 and 0.462 nM for MW07 (Fig. 1e). No cross-reactivity with SARS-CoV or MERS-CoV S1 recombinant proteins was detected for either mAb, as assessed using ELISA (Fig. 1f).

After its initial outbreak, the rapid spreading of SARS-CoV-2 has led to infections of millions of people and losses of many lives. Over the past few months, the spike protein of SARS-CoV-2 has been mutating[15]. To test whether MW05 and MW07 could cover currently spreading mutants, we next expressed RBD recombinant proteins from eight SARS-CoV-2 strains with reported high-frequency mutations. Binding assays showed that both MW05 and MW07 exhibited the same binding abilities to all RBD recombinant proteins. This result suggests that MW05 and MW07 may neutralize all eight of these strains (Fig. 1g and Supplementary Fig. 2).

**SARS-CoV-2 neutralization potency of MW05 and MW07.** To investigate the neutralizing activities of MW05 and MW07, we used in vitro assays to assess neutralization of first pseudovirus bearing the S protein of SARS-CoV-2 and then authentic virus. Both MW05 and MW07 inhibited pseudovirus infection of Huh7 cells effectively. Fifty percent neutralization titer (NT$_{50}$) was measured as 0.030 μg/ml for MW05 and 0.063 μg/ml for MW07 (Fig. 2a, b). We further evaluated the neutralizing activities of these two mAbs with authentic SARS-CoV-2 infection of Vero E6 cells. As expected, MW05 and MW07 blocked authentic SARS-CoV-2 entry into Vero E6 cells, with 100% NT (NT$_{100}$) around 1 μg/ml for MW05 and 5 μg/ml for MW07 (Fig. 2c, d). In summary, MW05 and MW07 exhibited substantial neutralization of both SARS-CoV-2 pseudovirus and authentic virus.

**FcγRIIB contributing to the ADE activity of MW05.** ADE has been observed for coronaviruses and several publications have shown that sera induced by SARS-CoV S protein enhanced viral entry into immune cells and inflammation[16–19]. To evaluate ADE activities of MW05 and MW07, we assessed the infection of SARS-CoV-2 pseudovirus and mAbs complex in THP-1, K562, and Raji cells. These cells are resistant to SARS-CoV-2 pseudovirus infection, as they do not express ACE2 receptor (Supplementary Fig. 3). Cells were incubated with the mixture of pseudovirus with serially diluted mAbs. Enhanced SARS-CoV-2 pseudovirus infection of Raji cells, but not of THP-1 or K562 cells, was observed for MW05 (Fig. 3a). Interestingly, no ADE activity was detected for MW07 on all three cell lines (Fig. 3b). Although MW05 and MW07 showed similar binding affinity to SARS-CoV-2 RBD recombinant protein and neutralization activity to SARS-CoV-2 pseudovirus, ADE activity was only observed for MW05. To explore whether antibody recognized epitopes correlate with the ADE activity of MW05 and MW07, we generated SARS-CoV-2 RBD mutants and measured their binding with MW05 and MW07. ELISA binding results showed that E484 and F490 are key amino acid residues for MW05 binding, while E484A or F490A mutation had no impact on the binding of MW07 (Supplementary Fig. 4). SARS-CoV-2 RBD and MW05 or MW07 Fab complexes were prepared for further crystal structure determination (Supplementary Fig. 5). These data indicate that MW05 and MW07 recognize different epitopes on RBD of SARS-CoV-2, which may explain the different ADE activities induced by these two mAbs. Next, we determined the FcγR expression profile of these three cell lines. FACS data revealed that Raji cells, which showed ADE activity for MW05, only express a relatively high level of FcγRIIB; THP-1 cells express high levels of FcγRIA and FcγRIIA; and K562 cells only express a high level of FcγRIIA (Fig. 3c). These results indicate

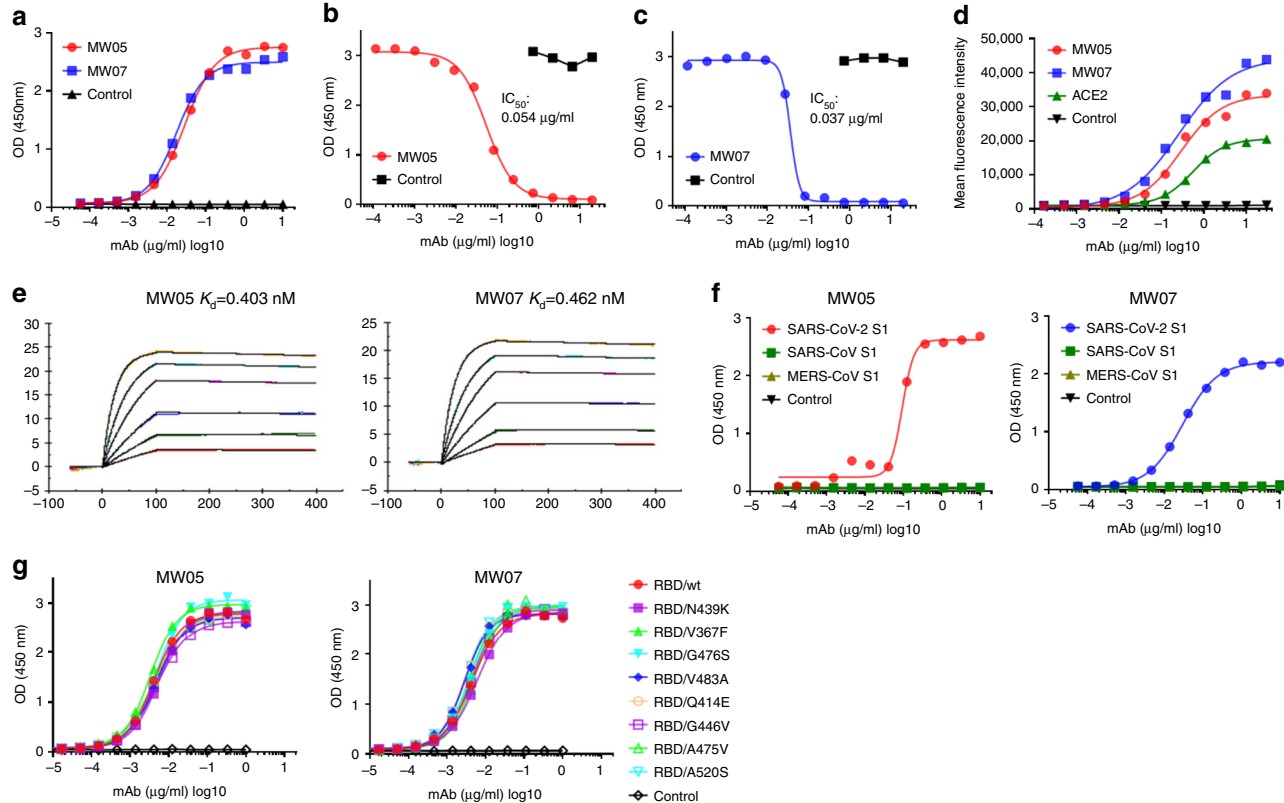

**Fig. 1 MW05 and MW07 disrupting the interaction of SARS-CoV-2 RBD with hACE2 receptor. a** The binding abilities of MW05 and MW07 to SARS-CoV-2 RBD recombinant protein were assessed by ELISA. **b**, **c** The abilities of MW05 and MW07 to block SARS-CoV-2 RBD interaction with ACE2 were evaluated by competition ELISA. **d** The binding of MW05 and MW07 to SARS-CoV-2 S protein expressed on HEK293 cells was measured by FACS. **e** The dissociation constants ($K_d$) of MW05 and MW07 to SARS-CoV-2 S1 recombinant protein were measured using a BIAcore S200 system. **f** The cross-reactivities of MW05 and MW07 to SARS-CoV-CoV-2, SARS-CoV, and MERS-CoV recombinant S1 subunit of spike proteins (S1) were tested by ELISA. **g** The binding of MW05 and MW07 to RBD recombinant proteins of SARS-CoV-2-mutated strains.

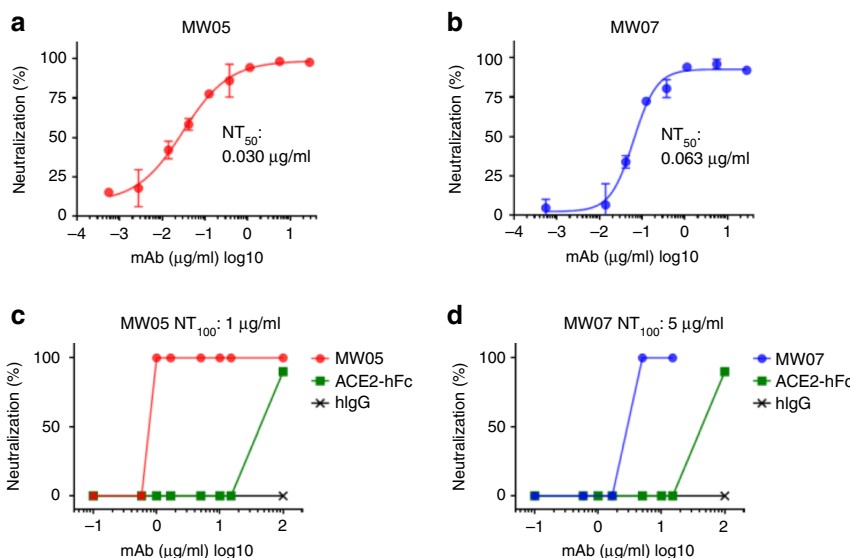

**Fig. 2 Neutralizing activities of MW05 and MW07. a**, **b** SARS-CoV-2 pseudovirus neutralizing activities of MW05 and MW07 were evaluated on Huh7 cells. Fifty percent neutralization titer ($NT_{50}$) was calculated by fitting the luciferase activities from serially diluted antibodies to a sigmoidal dose–response curve. The average ± SD from two independent experiments with technical duplicates is shown. **c**, **d** SARS-CoV-2 authentic virus-neutralizing activities of MW05 and MW07 were evaluated using Vero E6 cells. One hundred percent neutralization titer ($NT_{100}$) was labeled accordingly.

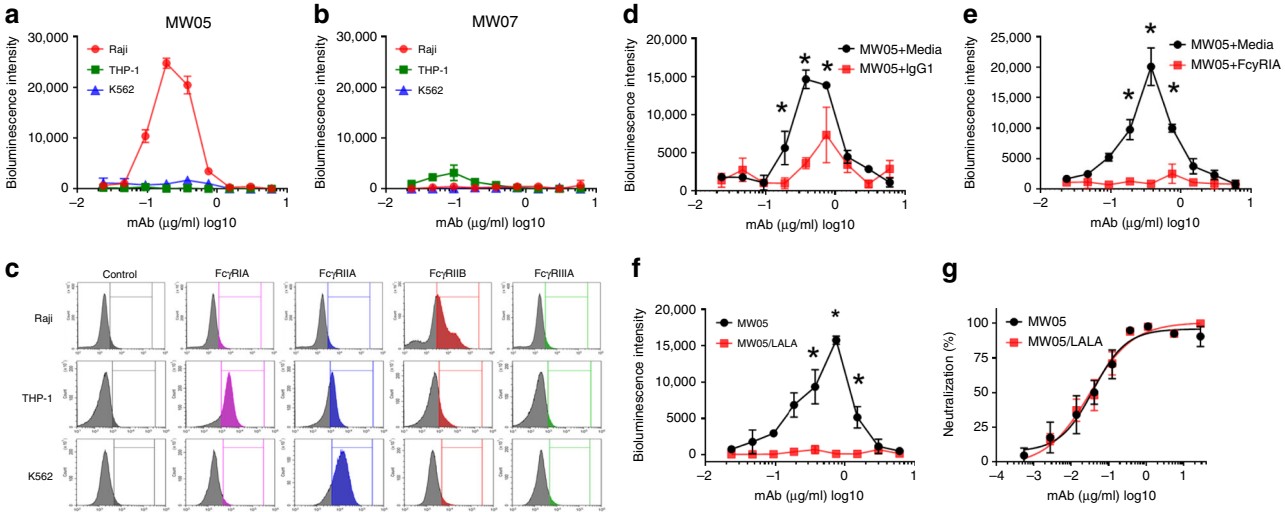

**Fig. 3 Crosslinking of Fc and FcγR contributing to ADE activities of MW05. a, b** ADE activities of MW05 and MW07 were assessed using SARS-CoV-2 pseudovirus. Pseudoviruses pre-incubated with serially diluted mAb mixtures were added to Raji, THP-1, and K562 cells to evaluate their ability to enhance infection. RPMI 1640 media containing 10% FBS was used as a negative control. **c** The expression level of FcγRs on Raji, THP-1, and K562 cells were checked by FACS. All cells except debris were gated and analyzed based on FSC-A/SSC-A plot. **d** ADE activities of MW05 on Raji cells pre-treated with media or 400 μg/ml irrelevant hIgG1 were assessed using SARS-CoV-2 pseudovirus. Statistical significance was calculated via unpaired two-tailed $t$-test. $*P < 0.05$. **e** ADE activities of MW05 pre-incubated with media or 80 μg/ml FcγRIA were assessed on Raji cells using SARS-CoV-2 pseudovirus. Statistical significance was calculated via unpaired two-tailed $t$-test. $*P < 0.05$. **f** The ADE activities of MW05 and MW05/LALA on Raji cells were compared using SARS-CoV-2 pseudovirus. Statistical significance was calculated via unpaired two-tailed $t$-test. $*P < 0.05$. **g** The pseudovirus neutralizing activities of MW05 and MW05/LALA on Huh7 cells were measured. The average ± SD from two independent experiments with technical duplicates is shown.

that FcγRIIB is the major FcγR contributing to the enhancement of SARS-CoV-2 infection mediated by MW05.

To further assess the ADE activity of MW05, we pre-incubated Raji cells along with irrelevant hIgG1 or MW05 along with FcγRIA recombinant protein to disrupt the interaction of MW05 Fc with FcγRIIB on Raji cells. Both pre-incubation strategies effectively inhibited the ADE activity of MW05 (Fig. 3d, e). FcγRIA has high affinity to the Fc of human IgG1. Accordingly, pre-incubation of MW05 with FcγRIA recombinant protein showed higher ADE inhibition than pre-incubation of irrelevant hIgG1 with Raji cells by disruption of MW05 Fc with FcγRIIB on Raji cells (Fig. 3d, e). To eliminate the risk of ADE and Fc-mediated acute lung injury in vivo, we introduced the LALA mutation to the Fc region of MW05 (MW05/LALA) to decrease the engagement of MW05 with FcγRs. This mutation completely eliminated ADE activity of MW05 without decreasing its neutralizing activity (Fig. 3f, g).

**Prophylactic and therapeutic efficacy of MW05/LALA in rhesus monkeys.** We evaluated the prophylactic and therapeutic efficacy of MW05/LALA in a rhesus monkey SARS-CoV-2 infection model. In the prophylactic (pre-challenge) group, three animals were injected intravenously with a single dose of MW05/LALA (20 mg/kg) one day before receiving $1 \times 10^5$ 50% tissue culture infectious dose ($TCID_{50}$) SARS-CoV-2 challenge via intratracheal inoculation (Fig. 4a). MW05/LALA antibody effectively protected animals from SARS-CoV-2 infection; almost no virus was detected in the oropharyngeal swabs of the prophylactic group (Fig. 4b).

In the treatment (post-challenge) group, three animals were first challenged with $1 \times 10^5$ $TCID_{50}$ SARS-CoV-2. Then, at day 1 post infection (dpi), a single dose of MW05/LALA (40 mg/kg) was administered intravenously to these animals (Fig. 4a). Animals in the control group ($n = 3$) were given a single dose of irrelevant hIgG1 (20 mg/kg) one day before virus challenge. In the control group, the viral loads in oropharyngeal swabs

increased to a peak of about $10^{7.0}$ RNA copies/ml on 4 dpi (Fig. 4b). Notably, virus titers decreased in the MW05/LALA treatment group immediately after administration. In the control group, viral titers at day 2 and day 3 are $10^{6.4}$ and $10^{6.2}$ copies/ml, respectively, while in the therapeutic group, viral titers decreased to $10^{4.8}$ and $10^{3.8}$ copies/ml, respectively. Almost no virus was detected in the MW05/LALA treatment group even on 4 dpi, the time point at which viral titers in the control group reached their peak. A single dose of MW05/LALA exhibited SARS-CoV-2 therapeutic efficacy in a rhesus monkey model, clearing virus in 3 days after antibody administration (Fig. 4b). No significant weight loss or body temperature change was observed in any of the animals during the study (Supplementary Figs. 6 and 7). Virus was only detected in the rectal swabs of two animals in the control group (Fig. 4c). No viral RNA was detected in nasal swabs or blood samples (Supplementary Fig. 8). Additionally, no significant abnormal hematology changes were observed (Supplementary Fig. 9).

Radiograph was taken and analyzed for all animals during the study. The lung X-ray images of the three experimental animals in the control group were clear on day 0 of the challenge, with no lung shadow observed. Ground glass-like shadows were observed in control animals C-2 and C-3 on 3 dpi and in the lungs of animal C-2 on 6 dpi. For all six animals of the prophylactic and therapeutic groups, X-ray images showed no significant lung shadows in any of the infected animals (Supplementary Fig. 10).

Rhesus monkeys challenged with SARS-CoV-2 were evaluated for tissue damage. One monkey from each group was euthanized for necropsy on 5, 6, and 7 dpi. Gross anatomy observations of lungs showed that the color of lungs of control animal C-1 sacrificed at 5 dpi appeared whitish pink, and the lobes were scattered with black spots visible to the naked eye. There was a hemorrhagic lesion (2 cm × 5 mm) at the edge of the right-lower lobe, a hemorrhagic lesion in the left-upper lobe, and a rust-colored lesion at the tip of left-middle lobe with a hard texture. On 6 dpi, the color of lungs of animal C-3 appeared whitish pink, and a local rust-colored lesion was observed at the tip margin of

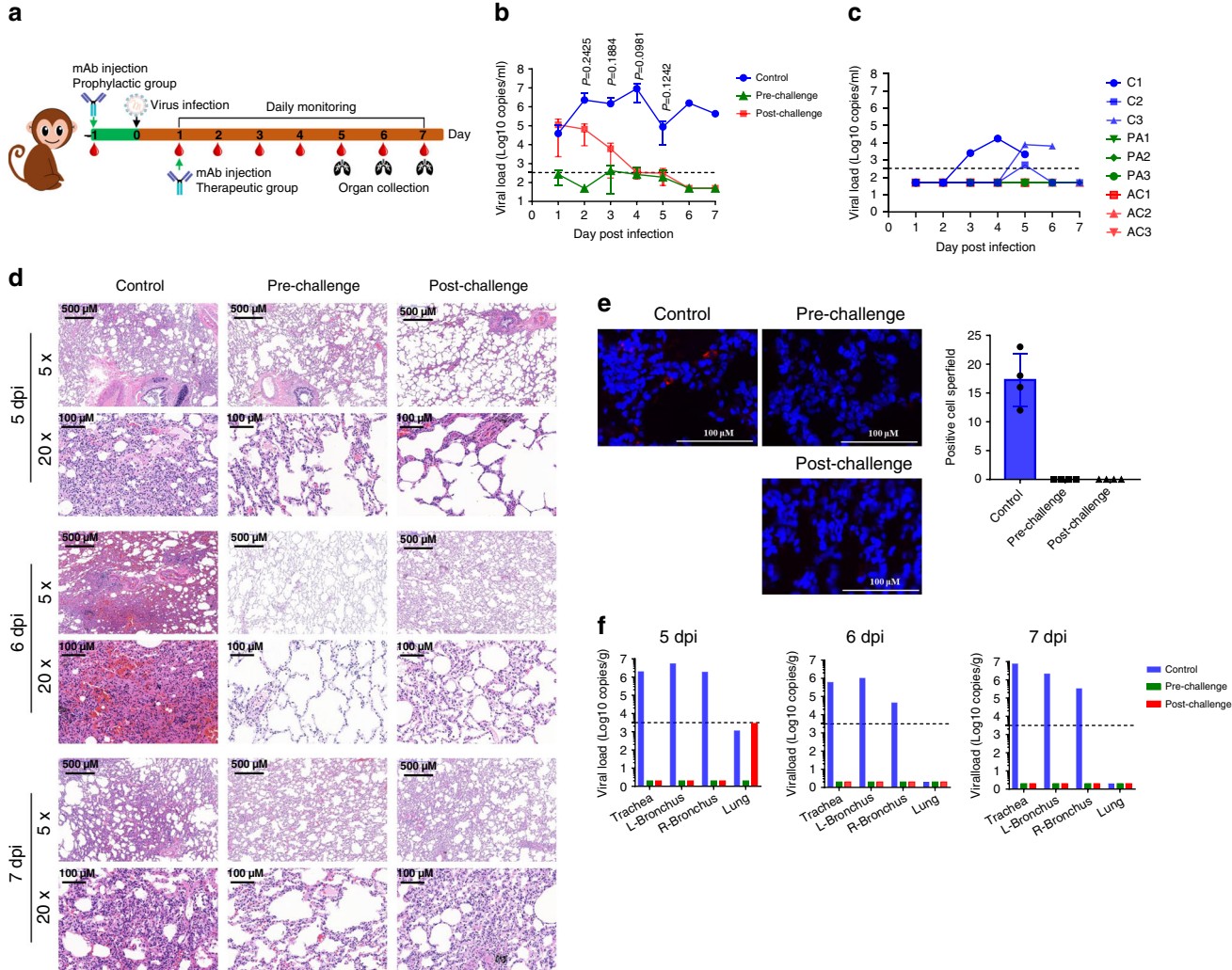

**Fig. 4 Prophylactic and therapeutic effects of MW05/LALA. a** A schematic of the experimental in vivo set up. Nine rhesus monkeys were divided into pre-challenge (prophylactic), post-challenge (therapeutic), and control groups with three animals in each group. Before virus challenge, the monkeys in the pre-challenge group were injected intravenously with a single dose of 20 mg/kg MW05/LALA. One day later, all monkeys were challenged with $1 \times 10^5$ $TCID_{50}$ SARS-CoV-2 via intratracheal intubation. A single dose of 40 mg/kg MW05/LALA was administered to each animal in the post-challenge group on day 1 post-challenge. Monkeys in the control group were given 20 mg/kg irrelevant hIgG1 one day before virus challenge. **b** Viral titers of oropharyngeal swabs at the indicated time points were evaluated using qRT-PCR. Data are average values from three monkeys ($n = 3$) for the first 5 days, from two monkeys ($n = 2$) for 6 dpi, and from one monkey ($n = 1$) for 7 dpi. The dotted line for limit of detection (200 copies/ml) is labeled. Statistical significance was calculated via unpaired two-tailed $t$-test. $P$ values at different time points are labeled. **c** Viral titer of rectal swabs at the indicated time points were evaluated by qRT-PCR. C indicates the control group, PA the pre-challenge group, and AC the post-challenge group. The dotted line for limit of detection (200 copies/ml) is labeled. **d** Histopathology and immunohistochemical examination of lung tissues from pre-challenge, post-challenge, and control monkeys. Two biological replicates were performed. **e** Immunofluorescence analysis of SARS-CoV-2 protein expression in lung tissues from pre-challenge, post-challenge, and control monkeys. Four fields were checked for each group. The average ± SD of $n = 4$ from two independent experiments is shown. Two biological replicates were performed. **f** Viral load analysis of trachea, bronchus, and lung tissues of experimental animals. L-Bronchus left bronchus, R-Bronchus right bronchus. The dotted line for limit of detection ($10^{3.5}$ copies/g) is labeled.

each lobe. There was a dark red hemorrhage lesion (2 cm × 4 mm) in the right-lower lobe, while no obvious abnormalities were observed in other areas. On 7 dpi, the lungs of animal C-2 appeared whitish pink, and there were scattered hemorrhage spots of different sizes in the left-lower lobe, right-upper lobe, right-middle lobe, and right-lower lobe. Dark pink hemorrhagic lesions were observed in the left- and right-lower lobe margins (Supplementary Fig. 11). The color of lungs of prophylactic animal PA-1 sacrificed at 5 dpi appeared pink, and no obvious abnormalities were observed in the gross necropsy. On 6 dpi, the lungs of animal PA-3 appeared pink; a rust-colored lesion was noted at the tip of the left-middle lobe with relatively hard

texture; a hemorrhage point was noted at the edge of the left-lower lobe. On 7 dpi, the lungs of animal PA-2 appeared pink; the lobes were scattered with macroscopically visible black spots; there was a white translucent membrane-like structure at the tip of the right-lower lobe, indicative of potential lung-pleural adhesion, while no other obvious abnormalities were observed (Supplementary Fig. 11). The color of lungs of therapeutic animal AC-1 sacrificed at 5 dpi appeared pink in color; the surface of the lobes were scattered with macroscopically visible congestion points, and no other obvious abnormalities were observed. On 6 dpi, the lungs of animal AC-3 appeared whitish pink, with visible black spots scattered on each lobe; a dark red (5 cm × 2 mm)

hemorrhagic lesion was observed on the medial side of the right-lower lobe, and no other obvious abnormalities were observed. On 7 dpi, the lungs of animal AC-2 appeared whitish pink, with rust-colored or bright-red hemorrhagic foci at the tips and margins of each lobe; adhesions were observed in the right-lower lobe, right-middle lobe, and right-upper lobe, and no other obvious abnormalities were observed (Supplementary Fig. 11).

Interstitial pneumonia symptoms were observed in the control group, including thickened alveolar septa, intensive infiltration of monocytes and lymphocytes, and proliferation of fibroblasts (Fig. 4d). We also observed fibrin exudation in some alveolar cavities, with the formation of hyaline membrane and pulmonary hemorrhaging (Fig. 4d). Monkeys in the treatment group displayed limited pathological lung changes, with overall alveolar structure intact and much lower levels of fibroblasts proliferation and leukocyte infiltration than that observed in the control monkeys (Fig. 4d). No lesions were observed in the lungs of the animal euthanized on 6 dpi and very mild pulmonary hemorrhaging of the animal euthanized on 5 and 7 dpi in the prophylactic group (Fig. 4d). Comparative analysis of pulmonary pathological changes in all animals in this study are shown in Supplementary Table 1. In summary, whether antibody was injected before or after virus challenge, MW05/LALA had a certain protective effect on alleviating the lung lesions caused by SARS-CoV-2 in experimental animals. The severity of damage caused by virus to the lungs was analyzed comprehensively through determination of the degree and extent of pulmonary pathological changes in different animals, which were determined specifically by analyzing the thickening of alveolar wall, inflammatory cell infiltration in alveolar wall, inflammatory cell infiltration and fibrosis in alveolar cavity, inflammatory cell infiltration around bronchioles and small vessels, etc. Control group animals showed typical pathological symptoms of interstitial pneumonia, including thickening of alveolar wall, fibroblast proliferation and fibrosis, inflammatory cell infiltration, alveolar edema and fibrin exudation, hyaline membrane formation, and pulmonary hemorrhage, etc. The pathological damage of lungs in the prophylactic and therapeutic groups were milder than that in the control group; no obvious alveolar edema or hemorrhage, or fibrin exudation or hyaline membrane formation was observed. However, there were individual differences in the pathological lesions among different animals. Lung lesions of some prophylactic and therapeutic group animals were not significantly improved, and the related effects and mechanisms need to be further studied.

Immunofluorescence analysis of the virus in lung tissues showed that SARS-CoV-2 protein has only been detected in the lung tissue of the control group on 5 dpi, but not 6 and 7 dpi. In comparison, viral protein was undetectable in the lung tissues of animals in the prophylactic and therapeutic groups (Fig. 4e). In order to further understand the distribution of SARS-CoV-2 in upper respiratory tract, trachea, bronchus, and lung tissues, samples were collected on 5, 6, and 7 dpi. Viral titers were then determined by quantitative reverse transcription PCR (qRT-PCR). Very low levels of viral RNA copies were detected in the lung tissues of control and therapeutic animals on 5 dpi. On 5, 6, and 7 dpi, high levels of SARS-CoV-2 RNA copies were detected in trachea and bronchi tissues of control animals, while no viral nucleic acid was detected from tissue samples of both prophylactic and therapeutic groups (Fig. 4f).

## Discussion

Animal models of SARS-CoV-2 infection and COVID-19 disease are still being developed. No single model has emerged as being more relevant for human diseases. By now, the rhesus macaques model is the best animal model to mimic various settings of human infection. The rhesus macaques model is widely used to assess efficacy of therapeutics and vaccines of SARS-CoV-2. In terms of the viral loads in respiratory samples, animals in the control group showed a quite similar level of viral loads as in publications of Munster et al.[20] and Doremalen et al.[21]. For the nasal swabs, a high viral load could be detected from Munster's paper mainly due to the combination inoculation route of intranasal, intratracheal, oral, and ocular. Pathology data in this study are also in line with previous studies. Animals in the control group showed evidence of lung injury by interstitial pneumonia, infiltration of mononuclear cells in the septa, and perivascular space. As the group sizes ($n = 3$) were small and large variations between monkeys were observed, more animal studies or clinical trials needed to be done to further confirm the prophylactic and therapeutic efficacy of MW05/LALA.

The global COVID-19 pandemic is running rampant over the world. There are great unmet medical needs for COVID-19 therapy as no SARS-CoV-2-specific drugs or vaccines have yet been approved. Neutralizing mAbs are promising agents to combat emerging infectious diseases. Our results showed the prophylactic and therapeutic efficacy of MW05/LALA on SARS-CoV-2 in vivo. This work paves the way for further development of antibody-based therapies for prophylactic and therapeutic treatment of COVID-19.

## Methods

**Ethics statement**. All neutralizing assays using SARS-CoV-2 authentic virus were performed in a biosafety level 3 (BSL-3) facility. Monkey studies were carried out in an animal biosafety level 4 (ABSL-4) facility with protocols approved by the Laboratory Animal Welfare and Ethics Committee of the Chinese Academy of Sciences. Informed consent was obtained from the COVID-19 patient before convalescent blood sample was collected.

**Cells and viruses**. HEK293 (ATCC, CRL-3216) cells, Huh7 (Institute of Basic Medical Sciences CAMS, 3111C0001CCC000679) cells, and Vero E6 (ATCC, CRL-1586) cells were cultured at 37 °C in Dulbecco's modified Eagle's medium (DMEM) supplemented with 10% fetal bovine serum (FBS). CHO-K1 (ATCC, CCL-61) cells were cultured at 37 °C in Dynamis Medium. Raji (ATCC, CCL-86) cells, THP-1 (ATCC, TIB-202) cells, and K562 (ATCC, CCL-243) cells were cultured at 37 °C in RPMI 1640 medium with 10% FBS. SARS-CoV-2 was isolated by the Center for Disease Control and Prevention of Zhejiang province. The virus strain used in animal study is SARS-CoV-2 (WIV04) (GISAID; accession number: EPI_ISL_402124). Vero E6 cells were applied to the reproduction of SARS-CoV-2 stocks.

**Recombinant protein generation**. The SARS-CoV-2 RBD (319-533aa, accession number: QHD43416.1), SARS-CoV-2 S1 (1-685aa, accession number: QHD43416.1), and SARS-CoV-2 RBD mutants recombinant proteins (Table 1) tagged with C-terminal 6× His were cloned into the pKN293E expression vector. HEK293 cells were transiently transfected with plasmids using 293fectinTM Transfection Reagent (Cat: 12347019, Life Technologies) when the cell density reached $1 \times 10^6$ cells/ml. Four days after transfection, the conditioned media was collected by centrifugation followed by purification using HisTrapTM HP (Cat: 17-5248-01, GE Healthcare). The purified protein was buffer exchanged into PBS using a Vivacon 500 concentrator (Cat: VS0122, Sartorius Stedim). For the generation of human ACE2-hFc and SARS-CoV-2 RBD-mFc recombinant proteins, RBD or ACE2 sequence (1–615aa, accession number: NP_068576.1) was cloned into mouse IgG1 or human IgG1 Fc backbone in pKN293E expression vectors and transiently transfected into HEK293 cells followed by media collection and purification using MabSelect SuRe antibody purification resin (Cat: 29-0491-04, GE Healthcare). SEC-HPLC and SDS-PAGE were used to check the size and purity of these recombinant proteins.

**Antibody discovery and expression**. mAbs were generated from SARS-CoV-2 RBD-specific memory B cells using single B cell isolation and cloning strategy[22]. For preparation of MW05 and MW07 recombinant antibodies, heavy- and light-chain plasmids were transiently co-transfected into HEK293 cells or stably expressed in CHO-K1 cells followed by purification with Protein A resin. Antibodies MW05/LALA and MW07/LALA were generated by introducing the LALA mutation (L234A and L235A) in the Fc region of IgG1 to abolish the binding with FcγRs and prepared as the same protocol used for generation of wild-type mAbs.

**Table. 1 Information of SARS-CoV-2 RBD mutants.**

| Mutants | Virus strain name | Accession ID | Data source |
|---|---|---|---|
| N439K | hCoV-19/Scotland/EDB162/2020 | EPI_ISL_425924 | GISAID |
| V367F | hCoV-19/England/20134027504/2020 | EPI_ISL_423136 | GISAID |
| G476S | hCoV-19/USA/WA-S28/2020 | EPI_ISL_417081 | GISAID |
| V483A | hCoV-19/USA/WA-S529/2020 | EPI_ISL_434289 | GISAID |
| Q414E | hCoV-19/USA/AZ-TGEN-TG268099/2020 | EPI_ISL_426500 | GISAID |
| G446V | hCoV-19/Australia/VIC329/2020 | EPI_ISL_426639 | GISAID |
| A475V | hCoV-19/USA/AZ-TGEN-TG268282/2020 | EPI_ISL_426504 | GISAID |
| A520S | hCoV-19/USA/WA_0432/2020 | EPI_ISL_426441 | GISAID |

**ELISA**. To assess the binding of mAbs to recombinant proteins (SARS-CoV-2 RBD, SARS-CoV-2 RBD mutants, SARS-CoV-2 S1, SARS-CoV S1 (Cat: 40150-V08B1, Sino Biological), MERS-CoV S1 (Cat: 40069-V08B1, Sino Biological)), recombinant proteins were first coated on 96-well ELISA plates at 1 μg/ml in 100 μl at 4 °C overnight. After blocking with 5% BSA in PBS, serially diluted mAbs were added to the plates and incubated for 60 min at 37 °C. Plates were washed and secondary Ab Goat Anti-Human IgG Fc-HRP (Cat: 109-035-098, Jackson ImmunoResearch) was added. TMB was used for color development and absorbance at 450 nm was measured using a microplate reader (Multiskan MK3, ThermoFisher). For the RBD/ACE2-hFc blocking assay, ACE2-hFc recombinant protein was coated on a 96-well ELISA plate at 0.75 μg/ml in 100 μl at 4 °C overnight. Equal volumes (100 μl + 100 μl) of pre-incubated RBD-mFc/mAb complex (RBD-mFc concentration: 100 ng/ml, mAb concentrations between 40 and 0.00023 μg/ml) were added to the plates and incubated for 60 min at 37 °C. Plates were washed and secondary Ab Goat Anti-Mouse IgG Fc-HRP (Cat:115-035-071, Jackson ImmunoResearch) was added. TMB was used for color development and absorbance at 450 nm was measured using a microplate reader.

**Flow cytometry assay**. The binding of MW05 and MW07 to S protein expression on cell surface was assessed by FACS. HEK293 cells were transiently transfected by SARS-CoV-2 Spike expression plasmid (Cat: VG40589-UT, Sino Biological) for 24–48 h. Cells were then collected and blocked with 5% BSA for 30 min at RT. Threefold serially diluted MW05, MW07, ACE2-hFc, and isotype control antibody were added into cells ($2 \times 10^5$ cells/sample in 100 μl) and incubated for 60 min on ice. After washing twice with 1× PBS, cells were stained with 1/200 diluted Goat Anti-human IgG Fc-FITC antibody (Cat: F9512, Sigma) for 45 min and analyzed using flow cytometry (CytoFLEX, Beckman Coulter). The FcγR expression profiles of Raji, THP-1, and K562 were determined by FACS. Cells were collected and washed twice with 1× PBS, and then blocked with Fc receptor blocking solution buffer (Cat: MX1505, Maokang Biological) for 30 min at RT. Then 10 μl anti-FcγRI antibody-FITC (Cat: 10256-R401-F, Sino Biological), anti-FcγRIIa antibody-FITC (Cat: 10374-MM02-F, Sino Biological), anti-FcγRIIIa antibody-FITC (Cat: 10389-MM41-F, Sino Biological), and FITC-labeled anti-FcγRIIb antibody (Cat: 398302, Biolegend) were added into cells ($1 \times 10^6$ cells/sample in 100 μl) and incubated for 60 min at 2–6 °C and analyzed using flow cytometry (CytoFLEX, Beckman Coulter). CytoExpert software was used for data analysis.

**Surface plasmon resonance**. SPR measurements were performed at room temperature using a BIAcore S200 system with CM4 biosensor chips (GE Healthcare). For all measurements, a buffer consisting of 150 mM NaCl, 10 mM HEPES, 3 mM EDTA, pH 7.4, and 0.005% (v/v) Tween-20 was used as running buffer. All proteins were exchanged into this buffer in advance. The blank channel of the chip served as the negative control. SARS-CoV-2 S1 recombinant protein was captured on the chip at 175 response units. Gradient concentrations of MW05 Fab or MW07 Fab (from 200 nM to 6.25 nM with twofold dilution) were then flowed over the chip surface. After each cycle, the sensor was regenerated with Gly-HCl (pH 1.5). The affinity was calculated using a 1:1 (Rmax Local fit) binding fit model with BIA evaluation software.

**Neutralization assay**. SARS-CoV-2 pseudovirus was prepared and provided by the Institute for Biological Product Control, National Institutes for Food and Drug Control (NIFDC)[23]. The $TCID_{50}$ was determined by the transduction of pseudovirus into Huh7 cells. For pseudovirus neutralization assay, 100 μl of mAbs at different concentrations were mixed with 50 μl supernatant containing 500 $TCID_{50}$ pseudovirus. The mixture was incubated for 60 min at 37 °C, supplied with 5% $CO_2$. All mAbs were tested in concentrations ranging from 0.55 ng/ml to 28 μg/ml in the context of Huh7 cells. One hundred microliters of Huh7 cell suspension ($2 \times 10^5$ cells/ml) were then added to the mixtures of pseudoviruses and mAbs for an additional 24 h incubation at 37 °C. Then, 150 μl of supernatant was removed, and 100 μl of luciferase detecting regents (Promega) was added to each well. After 2 min of incubation, each well was mixed 10 times by pipetting, and 150 μl of the mixture was transferred to a new microplate. Luciferase activity was measured using a

microplate luminometer (ThermoFisher). The $NT_{50}$ was calculated using Graph-Pad Prism 7.0. For SARS-CoV-2 authentic virus neutralization assay, Vero E6 cells were diluted and seeded into a 96-well plate with $1 \times 10^4$ cells/well in 100 μl volume at 37 °C. Sixteen hours later, cells were washed by 1× PBS for three times and added diluted antibodies in equal volume with the concentration ranging from 0.1 to 100 μg/ml. One hundred $TCID_{50}$ SARS-CoV-2 authentic virus was used for each well. Meanwhile, a control group without antibody was set up. A virus back-titration was performed to assess the correct virus titer used in each experiment. The cytopathic effect of each well was monitored every day and photographed at day 3 or day 4 after virus infection. All experiments were conducted following the standard operating procedures of the approved BSL-3 facility.

**ADE assay**. The ADE assays were performed using Raji, THP-1, and K562 cell lines. Twenty-five microliters of twofold serially diluted mAbs were mixed with 25 μL supernatant containing 250 $TCID_{50}$ pseudovirus. The mixture was incubated for 60 min at 37 °C, supplied with 5% $CO_2$. All mAbs were tested in the concentrations ranging from 6000 to 23.4 ng/ml. One hundred microliters of THP-1, Raji, and K562 cells at the density of $2 \times 10^6$ cells/ml were added to the mixtures of pseudoviruses and mAbs for an additional 24 h incubation. Then, the same volume of luciferase-detecting regents (Promega) was added to each well. After 2 min incubation, the luciferase activity was measured using a microplate luminometer (SpectraMax i3x, ThermoFisher).

**Animal experiments**. All animal experiments were performed according to the procedures approved by the Chinese Academy of Sciences and complied with all relevant ethical regulations regarding animal research. Nine 6- or 7-year-old rhesus monkeys (3 females and 6 males) were divided into three groups: a control group (one female and two males), a pre-exposure group (one female and two males), and a post-exposure group (one female and two males). The SARS-CoV-2 infection day was set as day 0. Rhesus monkeys in the control group were injected with 20 mg/kg negative control antibody. For the prophylactic study, monkeys in the pre-exposure group were given a single dose of 20 mg/kg MW05/LALA antibody intravenously one day before (day −1) being challenged with $1 \times 10^5$ $TCID_{50}$ SARS-CoV-2 via intratracheal routes. For the therapeutic study, monkeys in the post-exposure group were administrated with a single dose of 40 mg/kg MW05/LALA antibody intravenously one day after challenged with $1 \times 10^5$ $TCID_{50}$ SARS-CoV-2 via intratracheal routes. Oropharyngeal, nasal, rectal swabs, and blood samples were collected from day 1 to 7 post infection. Body weight and body temperature were monitored every day. Oropharyngeal, nasal, and rectal swabs were collected for 7 days. Blood samples were collected. White blood cells (WBC), neutrophils (NEUT), lymphocytes (LYMPH), and monocytes (MONO) were assessed for all monkeys from day 1 to 7 post infection. Swabs were placed into 1 ml of DMEM after collection. Viral RNA was extracted by the QIAamp Viral RNA Mini Kit (Qiagen) according to the manufacturer's instructions. RNA was eluted in 50 μl of elution buffer and used as the template for RT-PCR. The pairs of primers were used targeting S gene (Supplementary Table 2). Two microliters of RNA were used to verify the RNA quantity by HiScript® II One Step qRT-PCR SYBR® Green Kit (Vazyme Biotech Co., Ltd) according to the manufacturer's instructions. The amplification was performed as follows: 50 °C for 3 min, 95 °C for 30 s followed by 40 cycles consisting of 95 °C for 10 s, 60 °C for 30 s, and a default melting curve step in an ABI step-one machine.

**Histopathology and immunohistochemistry**. Animal necropsies were performed according to a standard protocol. The animals were euthanized on day 5, 6, and 7 post infection. The respiratory samples from each animal was collected for the viral load detection as well as the histological analysis. Samples for viral load detection from tissue were homogenized and used for viral RNA extraction based on the manufactory's instruction. Samples for histological examination were stored in 10% neutral-buffered formalin for 7 days, embedded in paraffin, sectioned, and stained with hematoxylin and eosin or Masson's trichrome prior to examination by light microscopy.

**Reporting summary**. Further information on research design is available in the Nature Research Reporting Summary linked to this article.

## Data availability

All relevant data are available in the article, Supplementary Information, or from the corresponding author X.G. upon reasonable request. Antibody and antibody sequences are available (by contacting X.G.) for research purposes only under an MTA, which allows the use of the antibody sequences for non-commercial purposes but not their disclosure to third parties. Source Data are provided with this paper.

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

## Acknowledgements

The rhesus macaque work was conducted at National Biosafety Laboratory, Wuhan, Chinese Academy of Sciences. We are particularly grateful to the running team of the laboratory for their work. We thank our colleagues from the Center for Instrumental Analysis and Metrology, Wuhan Institute of Virology, Chinese Academy of Sciences. We thank Dr. Zhou Yi-Wu for his assistance in the histopathological analysis. We are grateful to Dr. Dayong Tian and Dr. Qi An from Shanghai King-cell Biotechnology Co. Ltd. for helping us performing the SARS-CoV-2 authentic virus neutralizing assays in the P3 laboratory. We thank Ms. Hongyuan Ren and Mr. Pan Gao for their help in measuring the affinities in SPR assay. We thank Dr. Yu Mao and Ms. Wenlu Liang for the stimulating discussions. This work was supported by National Key R&D Program (2020YFC0848600).

## Author contributions

D.L., J.Z, X.G., and S.W. initiated and coordinated the project. R.W. and S.J. analyzed antigen sequences, expressed, and purified recombinant proteins. Z.L., M.W., and P.T. expressed and purified antibodies. R.W. and S.J. performed the binding and blocking assays. B.C. and W.J. performed the pseudovirus neutralization assays. C.G. and W.J. checked the expression profile of FcγRs. W.H. and L.W. designed and supervised the SARS-CoV-2 pseudovirus tests. X.G., B.C., W.J., and C.G. designed and performed the ADE experiments. G.L., A.W., and B.C supervised the protein quality control work. Z.Y., Y.P., C.S., X.H., and Y.Y. carried out the monkey studies with the help from H.Z., Y.C., and G.G. J.M. and Y.C. performed histopathology and immunohistochemistry assays. S.C performed the Sars-CoV-2 mutants sequence data analysis and design work. X.G., S.W., S.C., and J.Z. analyzed and prepared the manuscript with input from all authors.

## Competing interests

X.G., S.W., R.W., S.J., W.J., and C.G. are listed as inventors on the licensed patents for MW05 and MW07. S.W., R.W., S.J., M.W., W.J., Z.L., C.G., B.C., P.T., J.Z., X.G., and D.L. are employees of Mabwell (Shanghai) Bioscience Co., Ltd, and may hold shares in Mabwell (Shanghai) Bioscience Co., Ltd. The other authors declare no competing interests.
