## [Peer Review File · Nature Communications]

Reviewers' Comments:

Reviewer #1:

Remarks to the Author:

Wang and colleagues present two mAbs that (MW05 and MW07) have neutralizing activity against SARS-CoV-2 and does not cross-react with the S of SARS-CoV-1. Using an assay to measure ADE, they found that only MW05 can mediate internalization of pseudoparticles only in Raji cells and not in K562 or THP-1 cells. Further investigation find that FcγRIIB appears to be the determinant for Fc-FcγR dependent internalization of Raji cells. Of note, there is some concern over FcγR expression in THP-1 cells (discussed below). In vivo efficacy was validated in macaques where protection was found in both pre- and post-exposure prophylactic settings.

A lot of SARS-CoV-2 mAbs have already been published describing and the addition of two RBD-specific mAbs arguably adds little to the field. The idea of using cell lines to describe ADE is suggestive at best and may not reflect what may actually happen in nature. However, this should not be counted too much against the authors since the use of Raji cells have been used in CoV research. Of note, even in Raji cells SARS-CoV-1 replication was abortive (PMC3187504), meaning that while there is enhancement of entry, this mechanism did not lead to more viral progeny being made. This is in contrast to dengue virus, where it can replicate in FcγR-bearing cells and thus enhance disease to due increase viral replication. The finding that FcγRIIB mediated infection should be further discussed and authors should temper what this actually means in nature. In the dengue virus field, FcγRIIB has actually been demonstrated to inhibit ADE of infection (PMC3145677, PMC4673539).

The strength of this paper is the finding that they have two mAbs that can differentially engage FcR and that protection was validated in non-human primates. Specifically, the authors should focus on why they have 2 mAbs that can or cannot engage FcRs – this can significantly educate this nascent field. We currently do not know what are the determinants of ADE in SARS-CoV-2. It would be highly encouraging if the epitopes can be elucidated of these two mAbs as they can provide some clues as to the structural basis of the ADE phenomenon.

Line 70: Measuring RBD/ACE2 disrupting activities is not usually measured by a binding assay (ELISA), but rather through a neutralization assay, where the ability of an antibody to out compete for the viral receptor (ACE2) is measured. Please re-phrase the sentence.

Line 88: It might be better if the neutralization data for the wildtype SARS-CoV-2 presented as endpoint titers as the way the assay was performed has a binary readout.

Line 92: Authors should also cite Yip et al. (PMC3019510) and Jaume et al. (PMC3187504), as they observed an enhancement of infection of Raji cells. The use of FcγRI is noted and is actually a creative way to quench the IgG mAbs – nicely done. However, why didn't the authors just use the LALA mutants to do the ADE assay?

Line 95: It is interesting to observe that MW07 did not induce ADE, while MW05 did – especially since both appear to inhibit neutralization similarly. However, the significance of understanding the basis of determinants that affect ADE for this pandemic is paramount. It would be very beneficial for the field that the authors can identify the epitopes of both antibodies – through the generation of escape mutants.

Figure 3C: Authors report that THP-1 cells do not express FcγRIIB and does not support ADE of infection. However, prior work by Jaume et al., (PMC3187504) report that THP-1 cells are susceptible to ADE (to SARS-CoV-1) and do express R_γRII. The authors should discuss this discrepancy.

Reviewer #2:

Remarks to the Author:

The manuscript by Wang et al entitled "An antibody-dependent enhancement (ADE) activity eliminated neutralizing antibody with potent prophylactic and therapeutic efficacy against SARS-CoV-2 in rhesus monkeys" characterizes a monoclonal antibody against SARS-CoV-2 and its efficacy in challenged NHPs.

Overall the paper is well organized and addresses the urgent need for medical countermeasures against SARS-CoV-2. The in vitro characterization of the antibodies was complete and explained well. The authors went as far as to address the antibody-dependent enhancement seen in the use of some antibody therapies.

My main concern with the paper is the NHP efficacy study. Animals were treated with antibody 24 h before or 24 h after challenge constituting prophylactic and therapeutic treatment groups. The authors suggest the antibody is protective based on n=3 NHPs clearance of virus RNA in oropharyngeal swab at day 4 and no viral RNA above the level of detection in the prophylactic group. The differences in histopathology are based on an n=1 at days 6 and 7 post-challenge. The authors should discuss the limitations of this study and how it relates to similar work from other groups.

Specific concerns:

Minor

Line 77. The sentence "SARS-CoV-2 raced around the world after its initial outbreak." seems pedestrian and should be reworded using scientific terminology.

Line 77-78. The sentence "Over the past few months it has been mutating." should be more specific. What part of the viral genome is mutating? Also this statement requires a citation.

Line 127-128. The statement that viral titers decreased immediately after antibody administration requires more detail. When was the sample taken in regards to treatment (this should be stated in the materials and methods section)? How much was the decrease and was it statistically significant.

Line 136-137 States the one NHP from each group was euthanized on days 6 and 7 post-challenge. What happened to the third NHP in each group?

Lines 168 – 169, Methods, first paragraph last sentence should be edited for grammar and content.

Major

Figure 4B and C. What is the limit of detection for the assay. It is probably the dotted line but this should be specified.

Figure 4B. There should be some type of statistical analysis of this data.

Line 127-128. The statement that viral titers decreased immediately after antibody administration requires more detail. When was the sample taken in regards to treatment (this should be stated in the materials and methods section)? How much was the decrease and was it statistically significant.

Lines 148-146 State that SARS-CoV-2 protein only been detected in lung tissue of control group on 5 dpi but not on 6 dpi or 7 dpi. The data do not support this statement, there is no indication of animals being euthanized on 5 dpi. Also, the authors should state the number of viral positive cells per field and the number of fields viewed. The number of fields viewed should be stated for all groups.

Figure 4 E. State the day the samples were obtained. If this is Day 5, where is the other data from this day (viral load and histopathology)?

Where is the nasal swab data? I am assuming they were all negative but that should be stated in the results section.

The authors need to discuss why viral proteins were detected in the lungs but there was no genomic material detected. This seems counter intuitive as the PCR based assay should be more sensitive. The authors must include some discussion on how their results relate to similar studies in rhesus macaques, including the results of an in depth natural history study (Munster et al. 2020. Nature), protection provided by vaccination van (Doremalen et al. 2020 Nature) and antiviral study (Williamson

et al 2020 Nature).

The authors should discuss the limitations of the efficacy data, mainly the limited number of animals. Methods section, Animal experiments. The study design should be described in more detail including sampling times (swabs and blood collection) and euthanasia schedule.

Reviewer #3:

Remarks to the Author:

The authors describe the preclinical development of a potent neutralizing monoclonal antibody MW05, which was modified via a LALA mutation in the Fc portion to remove any ADE activity. They tested MW05/LALA both in a prophylactic as well as therapeutic setting in rhesus macaques. The authors provide interesting findings that are of high interest to the field. But I have a number of concerns, as expressed in my comments below, that need to be addressed.

Major comments:

- It is very surprising that the investigators, while able to find viral RNA in oropharyngeal swabs, and to a lesser extent rectal swabs, they were not able to find viral RNA in nasal swabs, although this is perhaps because they inoculated virus intratracheally. The researchers don't show any viral data on lower respiratory tract, such as bronchoalveolar lavages or tracheal washes during the live phase of the study. Were such samples not collected, or did they not give the desired results? In addition, at time of necropsy, even the control animals had no detectable viral RNA in lung tissues by RT-PCR, which is surprising considering that there is much virus in the trachea and bronchi, and that there also still histopathology. What was the limit of detection for the tissue RT-PCR methodology, as that is not mentioned in the manuscript.
- Figure 4. Did the authors use a scoring system for the gross pathology and histopathology? From other studies, it is known that even in SARS-CoV-2 infected control animals, the lesions are often spotty, as many regions may still be normal. How detailed and how systematic was the analysis performed to assure that all groups are evaluated the same way and the comparison is valid.
- Did the investigators take any radiographs of the animals during the study?
- To all figures where applicable, indicate what the error bars mean.
- In the animal study, as the group sizes (n=3) are small, was statistical analysis performed on the nasopharyngeal swabs (Fig. 4B)?

Minor comments:

- The figure legend of Figure 3 doesn't seem to match the figures (and probably reflects an earlier version of the figure), so it should be updated.
- As both MW05 and MW07 look promising in vitro, I am curious why the investigators selected MW05 for further development, especially as based on the in vitro data (Fig 3), MW07 didn't have any detectable ADE effects. Can the authors mention the reason?
- Line 118: Instead of "incubation", a better word would be "inoculation".
- Line 124: the text says that the controls received antibody 1 day post-inoculation, but the legend of figure 4 mentions it was given 1 day before virus inoculation. Please verify and correct.
- Line 140: The authors mention "cellulose". Do they mean a cellular reaction?
- The histology pictures in Figure 4 are too low resolution to interpret.
- Line 149 mentions detection of antigen in the lung tissue of the control group on day 5, but according to the rest of the manuscript, animals were euthanized either on day 6 or 7, so this is confusing. In Figure 4F, it is also not clear whether each panel presents the data on a single animal.
- What strain of SARS-CoV-2 was used for the inoculation of the animals?
- Line 284: Do body temperatures represent rectal temperatures, or was another method used?

Reviewer #1 (Remarks to the Author):

Wang and colleagues present two mAbs that (MW05 and MW07) have neutralizing activity against SARS-CoV-2 and does not cross-react with the S of SARS-CoV-1. Using an assay to measure ADE, they found that only MW05 can mediate internalization of pseudoparticles only in Raji cells and not in K562 or THP-1 cells. Further investigation find that FcγRIIB appears to be the determinant for Fc-FcR dependent internalization of Raji cells. Of note, there is some concern over FcR expression in THP-1 cells (discussed below). In vivo efficacy was validated in macaques where protection was found in both pre- and post-exposure prophylactic settings.

A lot of SARS-CoV-2 mAbs have already been published describing and the addition of two RBD-specific mAbs arguably adds little to the field. The idea of using cell lines to describe ADE is suggestive at best and may not reflect what may actually happen in nature. However, this should not be counted too much against the authors since the use of Raji cells have been used in CoV research. Of note, even in Raji cells SARS-CoV-1 replication was abortive (PMC3187504), meaning that while there is enhancement of entry, this mechanism did not lead to more viral progeny being made. This is in contrast to dengue virus, where it can replicate in FcR-bearing cells and thus enhance disease to due increase viral replication. The finding that FcγRIIB mediated infection should be further discussed and authors should temper what this actually means in nature. In the dengue virus field, FcγRIIB has actually been demonstrated to inhibit ADE of infection (PMC3145677, PMC4673539).

The strength of this paper is the finding that they have two mAbs that can differentially engage FcR and that protection was validated in non-human primates. Specifically, the authors should focus on why they have 2 mAbs that can or cannot engage FcRs- this can significantly educate this nascent field. We currently do not know what are the determinants of ADE in SARS-CoV-2. It would be highly encouraging if the epitopes can be elucidated of these two mAbs as they can provide some clues as to the structural basis of the ADE phenomenon.

Line 70: Measuring RBD/ACE2 disrupting activities is not usually measured by a binding assay (ELISA), but rather through a neutralization assay, where the ability of an antibody to out compete for the viral receptor (ACE2) is measured. Please re-phrase the sentence.

Response:

We appreciate the comments and suggestions of the reviewer. We agree that the neutralization assay is a better way to measure RBD/ACE2 disrupting activities of MW05 and MW07. The neutralization activities of MW05 and MW07 were measured in both SARS-CoV-2 pseudovirus and authentic virus systems (**Fig. 2**). We also performed the disrupting assay for MW05 and MW07 using RBD recombinant protein and ACE2 overexpressing HEK293 cells. Both MW05 and MW07 could effectively block the interaction of RBD with ACE2 (**Fig. R1**).

Fig. 2

Fig. R1. The abilities of MW05 and MW07 to block SARS-CoV-2 RBD interaction with ACE2 overexpressing HEK293 cells was evaluated by FACS. Pre-incubated mAb and SARS-CoV-2 RBD-mFc mixtures were added to ACE2 overexpressing HEK293 cells to check the binding of RBD to ACE2. Goat anti mouse IgG Fc-FITC was used as secondary antibody. Irrelevant human IgG1 was used as control.

Line 88: It might be better if the neutralization data for the wildtype SARS-CoV-2 presented as endpoint titers as the way the assay was performed has a binary readout.

Response:

Thanks for the suggestion.

Line 92: Authors should also cite Yip et al. (PMC3019510) and Jaume et al. (PMC3187504), as they observed an enhancement of infection of Raji cells. The use of FcγRI is noted and is actually a creative way to quench the IgG mAbs – nicely done. However, why didn't the authors just use the LALA mutants to do the ADE assay?

Response:

Thanks for the reminding. We cited these two publications (ref: 18 and 19). We conducted the ADE assay using LALA mutants. As shown in **Fig. 3F**, MW05/LALA

totally abolished the ADE activity compared with wild-type MW05.

Fig. 3F

Line 95: It is interesting to observe that MW07 did not induce ADE, while MW05 did – especially since both appear to inhibit neutralization similarly. However, the significance of understanding the basis of determinants that affect ADE for this pandemic is paramount. It would be very beneficial for the field that the authors can identify the epitopes of both antibodies – through the generation of escape mutants.

Response:

This is an insightful suggestion. MW05 and MW07 showed similar binding affinity to SARS-CoV-2 recombinant protein and neutralization activity to SARS-CoV-2 virus, but only MW05 enhanced the infection of SARS-CoV-2 to Raji cells. We generated several mutants to identified the epitope of these two mAbs. E484, Y489 and F490 on SARS-CoV-2 RBD are key amino acid residues contributing to MW05 binding (Fig. R2A). While, only Y489A mutation decreased the binding ability of MW07 (Fig. R2B). These results indicate that MW05 and MW07 recognize different epitopes. We believe that there is some correlation between MW05 recognized epitope and ADE activity.

We agree with the reviewer. It would be very beneficial for understanding the underlying mechanisms of ADE activity on SARS-CoV-2 by identification of MW05 and MW07 recognized epitopes. We also appreciate the reviewer’s suggestion by using escape mutation to identify the epitopes for MW05 and MW07.

To explore the detailed epitope information, we generated SARS-CoV-2 RBD, MW05 Fab and MW07 Fab recombinant proteins (Fig. R3). SARS-CoV-2 RBD and MW05 or MW07 Fab complexes were prepared for further crystal structure determination (Fig. R4).

Fig. R2. Binding of MW05 and MW07 to SARS-CoV-2 RBD mutants were measured

by ELISA.

Fig. R3. Generation and purification of SARS-CoV-2 RBD, MW05 Fab and MW07 Fab recombinant proteins.

Fig. R4. Generation and identification of SARS-CoV-2 RBD/MW05 Fab and SARS-CoV-2 RBD/MW07 Fab complexes.

Figure 3C: Authors report that THP-1 cells do not express FcγRIIb and does not support ADE of infection. However, prior work by Jaume et al., (PMC3187504) report that THP-1 cells are susceptible to ADE (to SARS-CoV-1) and do express RcγRII. The authors should discuss this discrepancy.

Response:

Thanks for the suggestion. In this paper (PMC3187504), Jaume and colleagues showed that THP-1 cells are susceptible to ADE for SARS-CoV-1. They also checked the expression profile of FcγRs using both RT-PCR and FACS assays. THP-1 cells only express RcyRIIA but not RcyRIIB (**Fig. 5A**, PMC3187504). This result is in line with our data showing in **Fig. 3C**. THP-1 cells express RcyRIIA but not RcyRIIB (**Fig. 3C**).

Fig. 3C

Reviewer #2 (Remarks to the Author):

The manuscript by Wang et al entitled “An antibody-dependent enhancement (ADE) activity eliminated neutralizing antibody with potent prophylactic and therapeutic efficacy against SARS-CoV-2 in rhesus monkeys” characterizes a monoclonal antibody against SARS-CoV-2 and its efficacy in challenged NHPs.

Overall the paper is well organized and addresses the urgent need for medical countermeasures against SARS-CoV-2. The in vitro characterization of the antibodies was complete and explained well. The authors went as far as to address the antibody-dependent enhancement seen in the use of some antibody therapies.

My main concern with the paper is the NHP efficacy study. Animals we treated with antibody 24 h before or 24 h after challenge constituting prophylactic and therapeutic treatment groups. The authors suggest the antibody is protective based on n=3 NHPs clearance of virus RNA in oropharyngeal swab at day 4 and no viral RNA above the level of detection in the prophylactic group. The differences in histopathology are based on an n=1 at days 6 and 7 post-challenge. The authors should discuss the limitations of this study and how it relates to similar work from other groups.

Specific concerns:

Minor

Line 77. The sentence “SARS-CoV-2 raced around the world after its initial outbreak.” seems pedestrian and should be reworded using scientific terminology.

Response:

We appreciate the comments and suggestions of the reviewer. We reworded the

sentence: “After its initial outbreak, the rapid spreading of SARS-CoV-2 has led to infections of millions of people and losses of many lives”. (Line 77-78).

Line 77-78. The sentence “Over the past few months it has been mutating.” should be more specific. What part of the viral genome is mutating? Also this statement requires a citation.

Response:

Thanks for the suggestion. We reworded the sentence to emphasize the mutations on SARS-CoV-2 spike protein and added a citation (ref: 15). “Over the past few months, the spike protein of SARS-CoV-2 has been mutating” (Line 78-79).

Line 127-128. The statement that viral titers decreased immediately after antibody administration requires more detail. When was the sample taken in regards to treatment (this should be stated in the materials and methods section)? How much was the decrease and was it statistically significant.

Response:

Thanks for your advice. The information was added in the methods section. Briefly, the SARS-CoV-2 infection day was set as day 0. For the prophylactic group, the antibody was injected on day -1 (one day before SARS-CoV-2 infection), while for the therapeutic group, the antibody was given on day 1 (one day after SARS-CoV-2 infection) as indicated in the schematic bar. The swabs were collected from day 1 to day 7 post infection.

In the therapeutic group, after antibody administration, viral titers decreased sharply compared with control group. In the control group, average viral titers at day 2, day 3 and day 4 are $10^{6.1}$ copies/ml, $10^{6.0}$ copies/ml and $10^{6.7}$ copies/ml, respectively. While, in the therapeutic group, average titers at day 2, day 3 and day 4 are $10^{4.0}$ copies/ml, $10^{3.3}$ copies/ml and $10^{2.5}$ copies/ml, respectively. As the group sizes (n=3) are small, even though viral RNA copies between control group and therapeutic (Post-challenge) group showed huge difference (100 to 10000 fold at different days), no significant difference was observed.

Line 136-137 States the one NHP from each group was euthanized on days 6 and 7 post-challenge. What happened to the third NHP in each group?

Response:

Animals were euthanized from each group on day 5, 6 and 7 post-challenge. We have added the animal data on day 5 in **Fig. 4D** and **Fig. 4F**.

Fig. 4

Lines 168 – 169, Methods, first paragraph last sentence should be edited for grammar and content.

Response:

We appreciate the reviewer’s careful reading. The sentence was modified: “Informed consent was obtained from the COVID-19 patient before convalescent blood sample was collected.” (Line 186-187)

Major

Figure 4B and C. What is the limit of detection for the assay. It is probably the dotted line but this should be specified.

Response:

The dotted line represented the limit of detection which is 200 copies/ml. We added in the figure legend.

Figure 4B. There should be some type of statistical analysis of this data.

Response:

Thanks for the suggestion. As the group sizes (n=3) are small, we used “t test” to analysis the nasopharygeal swabs at different points. Even though viral RNA copies between control group and post-challenge (therapeutic) group have huge difference (100 to 10000 fold from day 2 to day 4) , no significant difference was observed.

Line 127-128. The statement that viral titers decreased immediately after antibody administration requires more detail. When was the sample taken in regards to treatment (this should be stated in the materials and methods section)? How much was the decrease and was it statistically significant.

Response:

Thanks for the advice. The information was added in the Methods section. Briefly, the SARS-CoV-2 infection day was set as day 0. For the prophylactic group, the antibody was injected on day -1 (one day before SARS-CoV-2 infection), while for the therapeutic group, the antibody was given on day 1 (one day after SARS-CoV-2 infection) as indicated in the schematic bar. The swabs were collected from day 1 to day 7 post infection.

In the therapeutic group, after antibody administration, viral titers decreased sharply compared with control group. In the control group, average viral titers at day 2, day 3 and day 4 are $10^{6.1}$ copies/ml, $10^{6.0}$ copies/ml and $10^{6.7}$ copies/ml, respectively. While, in the therapeutic group, average titers at day 2, day 3 and day 4 are $10^{4.0}$ copies/ml, $10^{3.3}$ copies/ml and $10^{2.5}$ copies/ml, respectively. As the group sizes (n=3) are small, even though viral RNA copies between control group and therapeutic (Post-challenge) group showed huge difference (100 to 10000 fold at different days), no significant difference was observed.

Lines 148-146 State that SARS-CoV-2 protein only been detected in lung tissue of control group on 5 dpi but not on 6 dpi or 7 dpi. The data do not support this statement, there is no indication of animals being euthanized on 5 dpi. . Also, the authors should state the number of viral positive cells per field and the number of fields viewed. The number of fields viewed should be stated for all groups.

Response:

Sorry for the confusing. We did have euthanized the NHPs on day 5 post infection and updated the data in **Fig. 4**. About 10 positive cells in each field were observed and 5 fields were checked. We have added this information in figure legend.

Figure 4 E. State the day the samples were obtained. If this is Day 5, where is the other data from this day (viral load and histopathology)?

Response:

Yes, it's the data for day 5. The NHPs were euthanized on day 5, 6 and 7 post infection. We have added the viral load and histopathology data on day 5 post infection in **Fig. 4D and 4F**.

Fig. 4

Where is the nasal swab data? I am assuming they were all negative but that should be stated in the results section.

Response:

Yes, you are correct. No viral RNA was detected in nasal swabs for all groups (**Extended data Fig. 5A**). We stated this result in results section. (Line 137-138)

Extended data Fig. 5A

The authors need to discuss why viral proteins were detected in the lungs but there was no genomic material detected. This seems counter intuitive has the PCR based assay should be more sensitive.

Response:

Sorry for the confusing due to the viral load from lung on day 5 post infection was not presented in the previous manuscript. The viral proteins were only detected from lungs on day 5 post infection in control group. While, very low level of viral RNAs (below limit of detection, $\sim 10^{3.5}$ copies/g) were observed in the lungs from control and therapeutic animals (**Fig. 4F**).

Fig. 4F

The authors must include some discussion on how their results relate to similar studies in rhesus macaques, including the results of an in depth natural history study (Munster et al. 2020. Nature), protection provided by vaccination van (Doremalen et al. 2020 Nature) and antiviral study (Williamson et al 2020 Nature).

Response:

Thanks for the insightful suggestion. We have added some discussion in the revised manuscript. Animal models of SARS-CoV-2 infection and COVID-19 disease are still being developed. No single model has emerged as being more relevant for human diseases. By now, rhesus macaques model is the best animal model to mimic various settings of human infection. The rhesus macaques model is widely used to assess efficacy of therapeutics and vaccines of SARS-CoV-2.

In terms of the viral loads in respiratory samples, especially for throat swabs, the viral loads reached to about $10^{6.0}$ to $10^{7.0}$ copies/ml at day 2 post infection. Then, viral loads started to drop at day 5 post infection. Low level of viral RNAs were detected from rectal swabs of some monkeys in control group. No viral RNAs were detected from blood samples of all animals. No significant body temperature and body weight changes were observed. These data is in line with Munster's publication. For the nasal swabs, high viral load could be detected from Munster's paper mainly due to the combination inoculation route of intranasal, intratracheal, oral and ocular. Pathology data in our study is also in line with previous studies. Animals in the control group showed evidence of lung injury by interstitial pneumonia, infiltration of mononuclear cells in the septa and perivascular space.

As for the protection provided by vaccination (Doremalen et al. 2020 Nature), viral gRNAs were detected in nose swabs from all animals and no difference was found on any day between vaccinated animals and control animals. Viral RNAs were detected in a minority of animals, with no difference between groups. While in our study,

almost no viral RNAs were detected in MW05/LALA prophylactic group. In the control group, the viral loads reached to about $10^{6.1}$ copies/ml at day 2 and last to day 4 before dropping. For the viral RNA loads in lung tissues, viral RNAs were detected in 2 out of 6 animals in the prime-boost vaccinated group. While, no viral RNAs were detected in any lung tissues of MW05/LALA prophylactic animals.

The authors should discuss the limitations of the efficacy data, mainly the limited number of animals.

Response:

Thanks for your suggestion. We discussed in the revised manuscript about the limited number of animals.

Methods section, Animal experiments. The study design should be described in more detail including sampling times (swabs and blood collection) and euthanasia schedule.

Response:

Thanks for the advice. We have added more information about the animal study in the Methods section.

Reviewer #3 (Remarks to the Author):

The authors describe the preclinical development of a potent neutralizing monoclonal antibody MW05, which was modified via a LALA mutation in the Fc portion to remove any ADE activity. They tested MW05/LALA both in a prophylactic as well as therapeutic setting in rhesus macaques. The authors provide interesting findings that are of high interest to the field. But I have a number of concerns, as expressed in my comments below, that need to be addressed.

Major comments:

- It is very surprising that the investigators, while able to find viral RNA in oropharyngeal swabs, and to a lesser extent rectal swabs, they were not able to find viral RNA in nasal swabs, although this is perhaps because they inoculated virus intratracheally. The researchers don't show any viral data on lower respiratory tract, such as bronchoalveolar lavages or tracheal washes during the live phase of the study. Were such samples not collected, or did they not give the desired results? In addition, at time of necropsy, even the control animals had no detectable viral RNA in lung tissues by RT-PCR, which is surprising considering that there is much virus in the trachea and bronchi, and that there also still histopathology. What was the limit of detection for the tissue RT-PCR methodology, as that is not mentioned in the manuscript.

Response:

We appreciate reviewer's comments and deep insights. We share the same opinion with reviewer about the positive oropharyngeal swabs and "kind of negative" nasal swabs. And we are also curious about this phenotype. The possible reason is due to

the infection route. The viral RNA can always be detected from trachea and bronchus. The throat is physically close to the trachea. The virus could establish on-site replication in the throat while the nose still has the physical distance with trachea and the minor virus from breath cannot establish the on-site replication in the nose. This may cause the negative results from nasal swabs. Sorry, we regret that we didn't take fluid from either bronchoalveolar lavages or tracheal washes. The limit of detection for the tissues is between 10^3 to 10^4 copies/g based on the difference of the tissue weight. We have added this information in figure legend.

- Figure 4. Did the authors use a scoring system for the gross pathology and histopathology? From other studies, it is known that even in SARS-CoV-2 infected control animals, the lesions are often spotty, as many regions may still be normal. How detailed and how systematic was the analysis performed to assure that all groups are evaluated the same way and the comparison is valid.

Response:

That's a very good question. We agree with reviewer's comments. Based on other literature, the lesions cannot be distributed evenly. To have a better reflection about the pathogenesis of SARS-CoV-2 infection, we collected the samples from the upper, middle and lower position from the left and right lung lobes.

The scoring system:

1. A semi-quantitative score was performed based on the pathological level, including the thickness of the alveolar wall, the number of inflammatory cell infiltration, the alveolar cavity fibrosis and the inflammatory cells number and the degree of small bronchial disease. The scope of above pathological area was also evaluated. If the area is between 1% to 20%, it is mild level; the moderate level is between 21% to 50%; if the area is more than 50%, it will be defined as severe level.
2. All the slides were blind assessment by 3 pathologists.

- Did the investigators take any radiographs of the animals during the study?

Response:

Yes, we did. We took radiographs for all animals during the study. The X-ray of lung was performed on day 0, 3 and 6 post infection. Only one monkey (Control-3) in the control group showed reticular shadowing on 3 dpi and 6 dpi.

Control - 1

Control - 2

Control - 3

Post-Challenge - 1

Post-Challenge - 2

Post-Challenge - 3

- To all figures where applicable, indicate what the error bars mean.

Response:

Thanks for the suggestion. We updated the figure legends by adding statistical analysis information.

- In the animal study, as the group sizes (n=3) are small, was statistical analysis performed on the nasopharyngeal swabs (Fig 4B)?

Response:

Thanks for the suggestion. As the group sizes (n=3) are small, we used “t test” to analysis the nasopharyngeal swabs at different points. Even though viral RNA copies between control group and post-challenge (therapeutic) group have huge difference (100 to 10000 fold at different days), no significant difference was observed.

Minor comments:

- The figure legend of Figure 3 doesn’t seem to match the figures (and probably reflects an earlier version of the figure), so it should be updated.

Response:

Thanks for the reminding. We have updated the figure legend of Fig. 3.

- As both MW05 and MW07 look promising in vitro, I am curious why the investigators selected MW05 for further development, especially as based on the in vitro data (Fig 3), MW07 didn’t have any detectable ADE effects. Can the authors mention the reason?

Response:

MW05 and MW07 showed similar binding affinities to SARS-CoV-2 RBD

recombinant proteins and neutralization activities on SARS-CoV-2 pseudovirus system. However, in SARS-CoV-2 authentic virus system, MW05 showed about 5-fold higher neutralization potency than MW07. The 100% protection concentration for MW05 and MW07 are 1µg/ml and 5 µg/ml, respectively (**Fig. 2 C and D**). That's why we nominated MW05 as the lead antibody for further development.

- Line 118: Instead of “incubation”, a better word would be “inoculation”.

Response:

Thanks for the suggestion. We replaced the word “incubation” with “inoculation”.

- Line 124: the text says that the controls received antibody 1 day post-inoculation, but the legend of figure 4 mentions it was given 1 day before virus inoculation. Please verify and correct.

Response:

Sorry for the mistake. Animals in both control group and prophylactic group received antibody 1 day before virus challenge. The legend of Fig 4 was corrected. “Animals in the control group (n=3) were given a single dose of irrelevant hIgG1 (20 mg/kg) one day before virus challenge”. (Line 125-126)

- Line 140: The authors mention “cellulose”. Do they mean a cellular reaction?

Response:

No, that not mean a cellular reaction. To prevent the confusing in future, the word “fibrin” will be better to use it here. We change the “cellulose” to “fibrin” to describe the damage in the alveolar cavities.

- The histology pictures in Figure 4 are too low resolution to interpret.

Response:

Thanks for the reminding. We replaced the histology pictures in **Fig. 4** with high resolution ones.

Fig. 4D

- Line 149 mentions detection of antigen in the lung tissue of the control group on day 5, but according to the rest of the manuscript, animals were euthanized either on day 6 or 7, so this is confusing. In Figure 4F, it is also not clear whether each panel presents the data on a single animal.

Response:

Sorry for the confusing. We did euthanize the NHPs on day 5 post infection and necropsy was performed. The data in **Fig. 4F** represent single animal. The viral load

and histopathology data on day 5 post infection have been updated in **Fig. 4D and 4F**.

- What strain of SARS-CoV-2 was used for the inoculation of the animals?

Response:

The virus strain used in our study is SARS-CoV-2 (WIV04) (GISAID; accession number: EPI_ISL_402124). We have added the virus information in Methods section. (Line 194-195)

- Line 284: Do body temperatures represent rectal temperatures, or was another method used?

Response:

Yes. The body temperatures represent the rectal temperature.

Reviewers' Comments:

Reviewer #1:

Remarks to the Author:

Overall, the authors have addressed the most of the issues of the reviewers. However, there are some inconsistencies in the figures included in the response letter with the actual revised manuscript. The authors should address these edits first:

1. Where is Fig. R1 in the manuscript?
2. Extended Data Fig 1 in the response letter is actually Extended Data Fig. 3 in the revised manuscript. Please clarify.
3. The method by which the escape variants were generated should be described briefly in the Results section and more in detail in the Methods section.
4. It may benefit readers if the authors can generate a 3D model of the S protein highlighting the amino acids involved in mAb binding using their escape variant data.
5. Where are Fig. R2 and R3 in the revised manuscript. The authors mention that the RBD-Fab complexes were prepared for further crystal structure determination in Fig. R3, but Fig. R3 are only a western blot and purification of the complexes. Moreover, the right Western blot is mislabeled on the top of the gel – RBD is the Fab and MW07 Fab is the RBD. Please clarify.

Reviewer #2:

Remarks to the Author:

The authors are made all of the appropriate changes to the manuscript. It should be reviewed once more for grammar, but other than that I have no further comments.

Reviewer #3:

Remarks to the Author:

The authors have addressed the comments of the first review. There are still some spelling/grammar errors, for example:

line 121: duplication of "with"

line 149 ("radiographs were taken...")

And a number of times when the verb should be changed from plural to single or vice-versa.

Either the authors or journal editorial staff should go through this in detail.

Reviewer #1 (Remarks to the Author):

Overall, the authors have addressed the most of the issues of the reviewers. However, there are some inconsistencies in the figures included in the response letter with the actual revised manuscript. The authors should address these edits first:

1. Where is Fig. R1 in the manuscript?

Response:

Thank you for reviewing our manuscript and your valuable comments. We have moved Fig. R1 from the response letter to revised manuscript (Supplementary Fig. 1).

2. Extended Data Fig 1 in the response letter is actually Extended Data Fig. 3 in the revised manuscript. Please clarify.

Response:

Thanks for the reminding. It is Supplementary Fig. 4 in the revised manuscript.

3. The method by which the escape variants were generated should be described briefly in the Results section and more in detail in the Methods section.

Response:

Thanks for the suggestion. We have added some description in the Results section.

4. It may benefit readers if the authors can generate a 3D model of the S protein highlighting the amino acids involved in mAb binding using their escape variant data.

Response:

Thanks for the suggestion. We have added this data to Supplementary Fig. 4.

Supplementary Fig. 4

5. Where are Fig. R2 and R3 in the revised manuscript. The authors mention that the RBD-Fab complexes were prepared for further crystal structure determination in Fig. R3, but Fig. R3 are only a western blot and purification of the complexes. Moreover, the right Western blot is mislabeled on the top of the gel – RBD is the Fab and MW07 Fab is the RBD. Please clarify.

Response:

We have moved Fig. R2 and R3 from the response letter to revised manuscript (Supplementary Fig. 5). We generated the RBD/Fab complex, and Supplementary Fig. 5a and 5c are the characterization of RBD/Fab complexes. Supplementary Fig. 5b and 5d are SDS-PAGE of the complexes. Thanks for your careful reading. We have corrected the label in Supplementary Fig. 5d.

Supplementary Fig. 5

Reviewer #2 (Remarks to the Author):

The authors are made all of the appropriate changes to the manuscript. It should be reviewed once more for grammar, but other than that I have no further comments.

Response:

Thank you for reviewing.

Reviewer #3 (Remarks to the Author):

The authors have addressed the comments of the first review. There are still some spelling/grammar errors, for example:

line 121: duplication of "with"

line 149 ("radiographs were taken...")

And a number of times when the verb should be changed from plural to single or vice-versa.

Either the authors or journal editorial staff should go through this in detail.

Response:

Thanks for your valuable suggestions. We have corrected them in revised manuscript.